# Cell Voltage Equalizer Using a Selective Voltage Multiplier with a Reduced Selection Switch Count for Series-Connected Energy Storage Cells

**Masatoshi Uno *** , **Teruhisa Ueno and Koji Yoshino**

College of Engineering, Ibaraki University, Hitachi 316-8511, Japan; 19nm609g@vc.ibaraki.ac.jp (T.U.); 18nm658l@vc.ibaraki.ac.jp (K.Y.)
*   Correspondence: masatoshi.uno.ee@vc.ibaraki.ac.jp; Tel.: +81-294-38-5098

**Abstract:** Cell voltage equalization is mandatory to eliminate voltage imbalance of series-connected energy storage cells, such as lithium-ion batteries (LIBs) and electric double-layer capacitors (EDLCs), to ensure years of safe operations. Although a variety of cell equalizers using selection switches have been proposed, conventional techniques require numerous switches in proportion to the cell count and are prone to complexity. This paper proposes a novel cell voltage equalizer using a selective voltage multiplier. By embedding selection switches into the voltage multiplier-based cell voltage equalizer, the number of selection switches can be reduced in comparison with that in conventional topologies, realizing the simplified circuit. A prototype for twelve cells was built, and an equalization test using LIBs was performed. The voltage imbalance decreased down to approximately 20 mV by the proposed equalizer, and the standard deviation of cell voltages at the end of the equalization test was as low as 10 mV, demonstrating its equalization performance.

**Keywords:** electric double-layer capacitor (EDLC); equalization; lithium-ion battery (LIB); selection switch; voltage imbalance

---

## 1. Introduction

In general, voltages of series-connected energy storage cells, such as lithium-ion batteries (LIBs) and electric double-layer capacitors (EDLCs), gradually become imbalanced due to nonuniform individual cell characteristics in capacitance, internal impedance, coulombic efficiency, and self-discharge rate. Temperature mismatch in a battery pack or module also leads to the occurrence of voltage imbalance because the self-discharge rate is dependent on temperature—the higher the temperature, the faster the self-discharge will be [1]. Some cells in a voltage-mismatched pack might be overcharged and -discharged due to voltage imbalance during charging and discharging processes, respectively. Charging–discharging energy storage cells beyond safety boundaries likely results in premature degradation and hazardous situations of fire or, in the worst case, an explosion. Thus, cell voltage equalization is mandatory to eliminate the voltage imbalance to prevent operations beyond safety boundaries [2].

Various kinds of cell voltage equalizers have been proposed and commercialized. Adjacent cell-to-cell equalizers based on non-isolated bidirectional converters, such as PWM converters [3,4] and switched capacitor converters [5–8], are the most straightforward approach for cell equalization. However, in addition to a large number of converters necessary for adjacent cell-to-cell equalization architectures, energy transfer is limited only between neighboring cells, collectively increasing power conversion loss in the course of equalization, especially in large-scale systems comprising numerous cells connected in series.

On the other hand, pack-to-cell equalizers, which are based on a single-input–multi-output converter, can achieve reduced numbers of converters and active switches [9–20]. A conventional pack-to-cell equalizer based on a voltage multiplier is shown in Figure 1 as an example. This topology requires only two switches, regardless of cell count, achieving a simplified circuit. This equalizer automatically supplies an equalization current toward the least charged cell having the lowest voltage in the pack, realizing the automatic equalization even without feedback control. However, this automatic equalization cannot be simply applied to LIBs because the relatively large voltage drop across internal impedances of LIBs hinders and slows down the voltage equalization process.

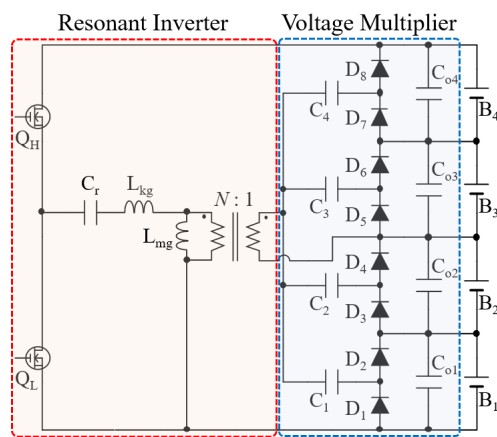

**Figure 1.** Conventional equalizer using an inverter and voltage multiplier [19].

Meanwhile, cell-to-cell equalizers using cell selectin switches have been intensively developed for EV battery systems [21–35]. Target cells having the lowest or highest voltages in the pack are selected by the cell selection switches so that their stored energies are exchanged to equalize cell voltages, regardless of voltage drops across internal impedance. Typical equalization systems using cell selection switches are listed in Figure 2. In general, selection switches are bidirectional switches consisting of two metal-oxide-semiconductor field-effect transistors (MOSFETs) connected back-to-back in order to block bidirectional current flow. The direct cell-to-cell equalizer using a unidirectional isolated converter (e.g., a flyback converter), as shown in Figure 2a, requires $4n$ selection switches, and therefore, the system is prone to complexity [21,22]. Needless to say, a unidirectional switch is also necessary for the isolated converter. With the pack-to-cell or cell-to-pack equalizers with selection switches (Figure 2b) [23–26], the numbers of selection switches can be halved (i.e., $2n$), but there is still a room for improvement. The pack-to-cell equalizer with polarity switches (Figure 2c) can further reduce the switch count as low as $n + 5$ [27]. The direct cell-to-cell equalizer with an energy storage medium (Figure 2d), such as a capacitor, inductor, resonant tank, etc., can reduce the switch count as low as $n + 1$ [28–35], but selection switches in many existing direct cell-to-cell equalizer must operate at a high switching frequency, for which numerous high-frequency gate drivers are also indispensable. Furthermore, four unidirectional switches are also necessary in the case of the topology in Figure 2d. Since each bidirectional selection switch and unidirectional switch requires a gate driver as well as its auxiliary power supply, the switch count can be an index to represents the circuit complexity. The number of selection switches should desirably be reduced as much as possible to simplify the circuit and to reduce the cost.

This paper proposes a novel pack-to-cell equalizer using a selective voltage multiplier. By embedding cell selection switches into a voltage multiplier-based cell voltage equalizer, the numbers of selection switches can be reduced to $n$, achieving the simplified circuit. Section 2 describes the proposed equalizer and its major features, followed by the operation analysis in Section 3. The experimental results of an equalization test for twelve LIB cells connected in series are shown in Section 4.

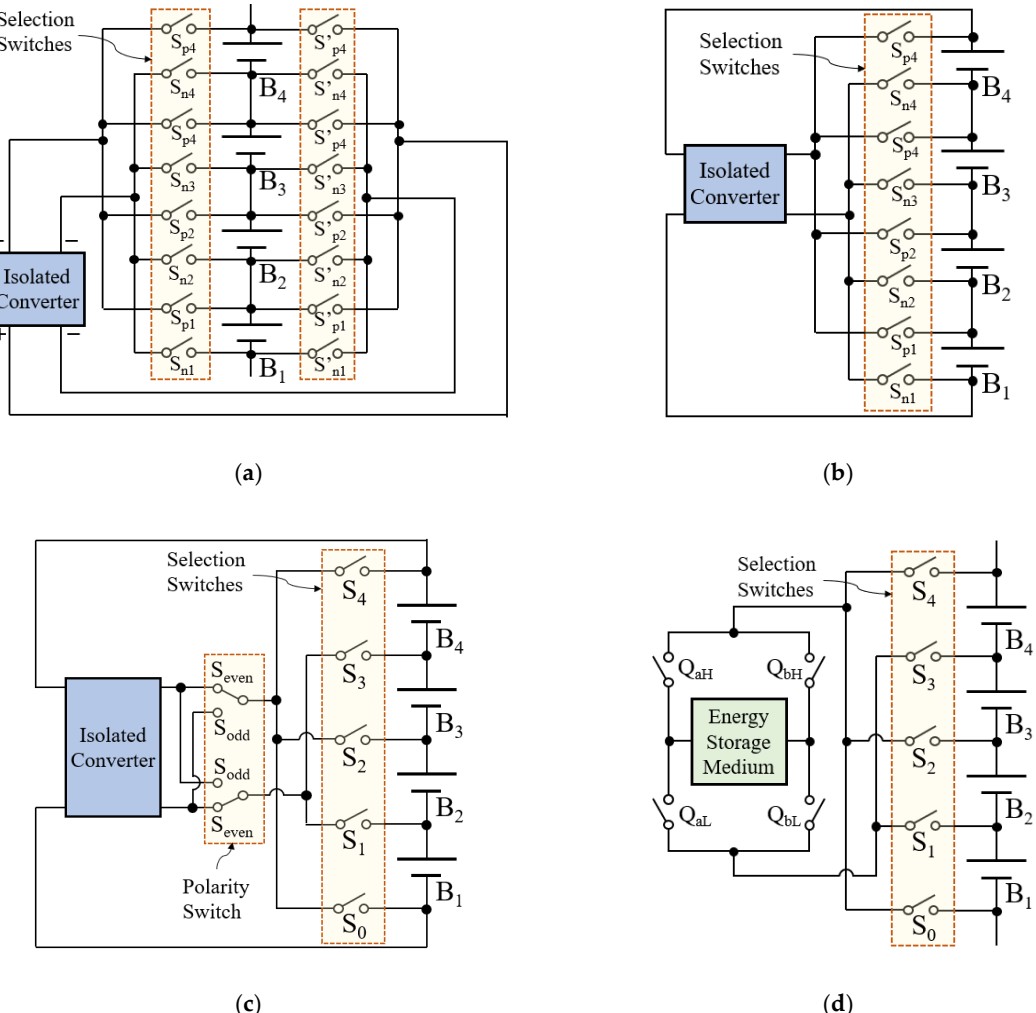

**Figure 2.** Conventional cell equalizers with selection switches: (**a**) direct cell-to-cell equalizer based on an isolated converter; (**b**) pack-to-cell equalizer; (**c**) pack-to-cell equalizer with polarity switches; and (**d**) direct cell-to-cell equalizer with energy storage medium.

## 2. Proposed Cell Voltage Equalizer Using Selective Voltage Multiplier

### 2.1. Topology

The proposed voltage equalizer using the selective voltage multiplier for four cells connected in series is shown in Figure 3 as an example. This equalizer consists of the resonant inverter and voltage equalizer with selection switches embedded. In comparison with the conventional pack-to-cell equalizer using the voltage multiplier (see Figure 1), coupling capacitors $C_1$–$C_4$ are replaced with selection switches $S_1$–$S_4$. $S_1$–$S_4$ are bidirectional switches comprising two MOSFETs connected back-to-back, as shown in the inset of Figure 3. The symmetric half-bridge resonant inverter is employed, but its fundamental operation principle is identical to that of the conventional equalizer shown in Figure 1.

High- and low-side switches, $Q_H$ and $Q_L$, are alternately driven in a complementary mode with a fixed 50% duty cycle to generate resonant AC current for the transformer secondary side. A selection switch corresponding to the least charged cell is activated, and the AC current transferred from the resonant inverter is rectified by the voltage multiplier, producing a DC equalization current for the least charged cell. The detailed operation principle is discussed in Section 3.

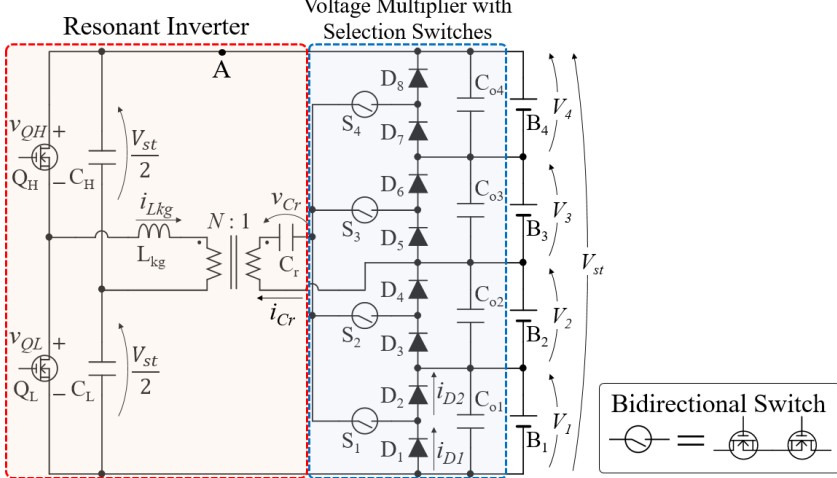

**Figure 3.** Proposed equalizer with selective voltage multiplier for four cells connected in series.

*2.2. Features*

Conventional selection switch-based equalizers require at least $n + 1$ bidirectional selection switches, as introduced in Section 1. The topology in Figure 2d comprises the least number of selection switches (i.e., $n + 1$), but it requires four unidirectional switches. With the proposed equalizer, on the other hand, the number of bidirectional selection switches and unidirectional switches can be reduced to as low as $n$ and two, respectively, reducing the circuit complexity and cost. Furthermore, the selection switches in the proposed equalizer do not need to operate at a high frequency, and therefore, gate driver circuits for selection switches can be simpler and less powerful than those needed in conventional cell-to-cell equalizer using selection switches [28–35]. Although diodes $D_1$–$D_8$ and smoothing capacitors $C_{o1}$–$C_{o4}$ are additionally necessary for the voltage multiplier, these are passive components and do not need auxiliary circuits. Hence, the added complexity due to the passive components in the voltage multiplier would be minor compared to selection switches, for which auxiliary circuits, including gate drivers and power supplies, are indispensable.

The resonant inverter in the proposed equalizer is essentially identical to that in the convention equalizer shown in Figure 1 [19]. It operates in a discontinuous conduction mode (DCM), by which an equalization current supplied to cell(s) can be automatically constant even without feedback control. This inherent constant current characteristic is a suitable feature for energy storage devices because LIBs and EDLCs are essentially a voltage source.

## 3. Operation Analysis

*3.1. Operation Modes*

The operation analysis is performed for the case that $B_1$ is the least charged cell in the battery pack and $S_1$ is activated. All circuit elements are assumed ideal unless otherwise noted. Theoretical key operation waveforms and current flow paths are shown in Figures 4 and 5, respectively.

Mode 1 ($0 \leq t < T_1$) (Figure 5a): The gating signal for $Q_H$, $v_{GS.H}$, is applied to turn on $Q_H$, achieving zero current switching (ZCS) turn-on. $L_{kg}$ and $C_r$ start resonating, and the current of $C_r$, $i_{Cr}$, sinusoidally changes. On the transformer secondary side, the resonant current flows through the activated selection switch of $S_1$ and the high-side diode corresponding to $B_1$, $D_2$. This operation mode lasts until $i_{Cr}$ becomes zero.

Mode 2 ($T_1 \leq t < T_2$) (Figure 5b): $Q_H$ and $Q_L$ are still on and off, respectively. The polarity of $i_{Cr}$ is reversed, while the low-side diode corresponding to $B_1$, $D_1$, starts to conduct. $v_{GS.H}$ is removed before $i_{Cr}$ comes back to zero in order to turn off $Q_H$ at zero voltage switching (ZVS). At the same time, the body diode of $Q_H$ conducts. This operation mode ends when $i_{Cr}$ becomes zero.

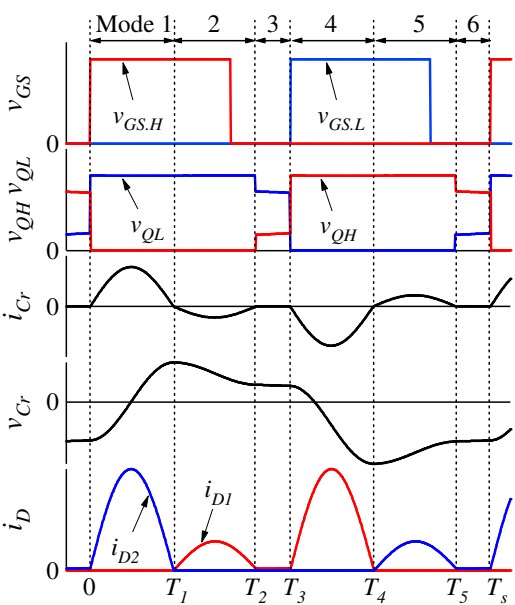

**Figure 4.** Theoretical waveforms when $B_1$ is the least charged cell.

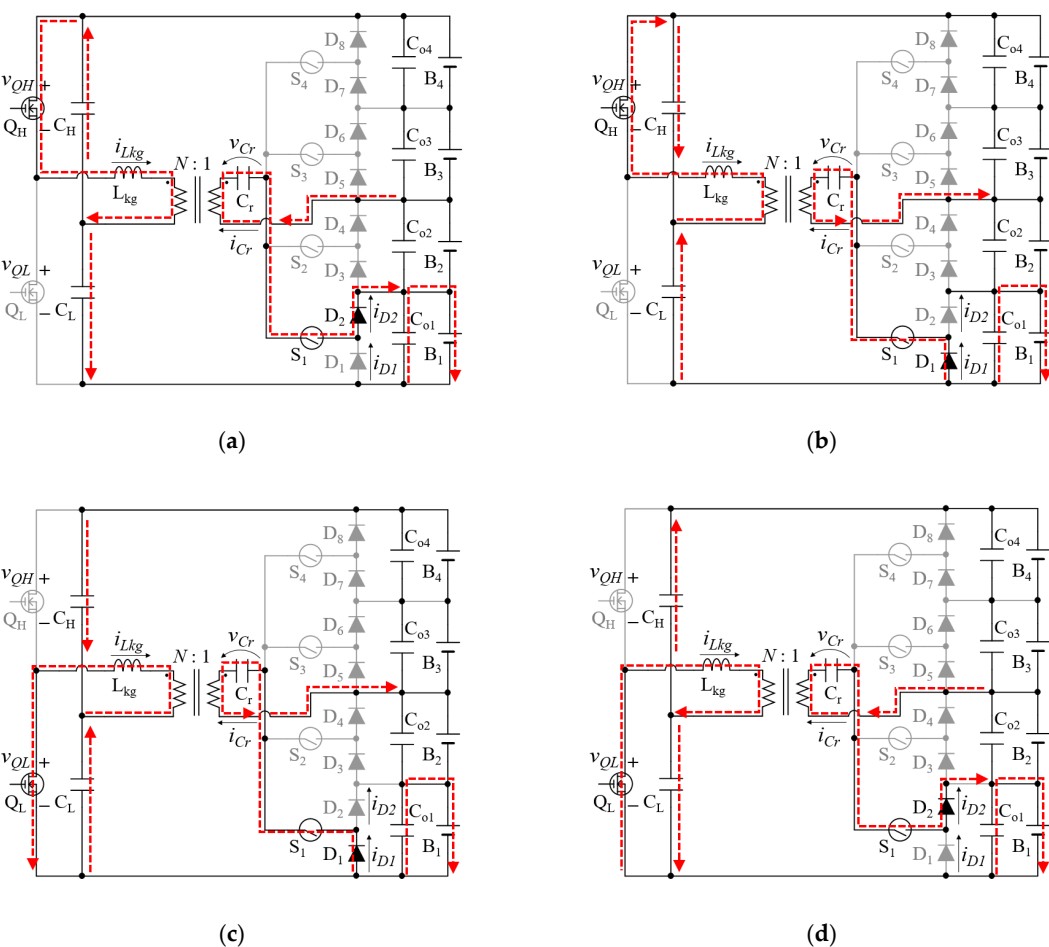

**Figure 5.** Current flow paths: (**a**) Mode 1; (**b**) Mode 2; (**c**) Mode 4; (**d**) Mode 5.

Mode 3 ($T_2 \leq t < T_3$) (not shown): This mode is unique to the DCM operation. No currents flow in this mode, except for the smoothing capacitors.

Mode 4 ($T_3 \leq t < T_4$) (Figure 5c): The gating signal for $Q_L$, $v_{GS.L}$ is applied to turn on $Q_L$ at ZCS. $L_{kg}$ and $C_r$ start resonating again. $D_1$ on the secondary side conducts. This operation mode ends when $i_{Cr}$ becomes zero.

Mode 5 ($T_4 \leq t < T_5$) (Figure 5d): $Q_H$ and $Q_L$ are still off and on, respectively. The polarity of $i_{Cr}$ is opposite to that in Mode 4. The current on the secondary side flows through $D_2$. $v_{GS.L}$ is removed before $i_{Cr}$ comes back to zero so as to turn off $Q_L$ at ZVS. As $i_{Cr}$ becomes zero, the operation moves to Mode 6.

Mode 6 ($T_5 \leq t < T_s$) (not shown): This mode is identical to Mode 3, and no currents flow in the circuit.

In summary, the resonant current flows through the activated selection switch ($S_1$) and diodes connected to the least charged cell. The AC current is rectified by the diodes in the voltage multiplier, and a DC equalization current is supplied to the least charged cell only.

### 3.2. Operation Boundary and Equalization Current

As can be seen in Figure 4, half the switching period ($0.5T_s$) must contain a full resonant period $T_r$. Hence, the following operation boundary needs to be fulfilled:

$$2f_S \leq f_r, \tag{1}$$

where $f_s$ is the switching frequency and $f_r$ is the resonant frequency.

Since the resonant inverter in the proposed equalizer is identical to that in the conventional pack-to-cell equalizer [19] (see Figure 1), the equalization current supplied to the least charged cell can be expressed in the identical form, as:

$$I_{eq} \approx \frac{2N\omega_s V_{in}}{\pi Z_0 \omega_r}, \tag{2}$$

where $\omega_s$ and $\omega_r$ are the angular switching and resonant frequencies, respectively, $Z_0$ is the characteristic impedance of the resonant tank:

$$Z_0 = \sqrt{\frac{L_{kg}N^2}{C_r}}, \omega_r = \sqrt{\omega_0^2 - \gamma^2}, \tag{3}$$

where $\omega_0$ is the characteristic angular frequency, and $\gamma$ is the damping factor given by:

$$\omega_0 = \frac{N}{\sqrt{L_{kg}C_r}}, \gamma = \frac{R}{2L_{kg}}, \tag{4}$$

where $R$ is the sum of resistive components in the resonant current path.

The equalization current $I_{eq}$ is independent on cell voltage, as Equation (2) does not contain the cell voltage. By properly designing the resonant tank parameters, currents in the circuit can be limited within desired levels even without feedback control.

### 3.3. Equalization Algorithm

The equalization algorithm for the proposed equalizer is shown in the form of the flow chart in Figure 6. At the beginning, all the selection switches are off. Cell voltages are measured, and open-circuit voltages $V_{OC}$ are estimated by compensating a voltage drop across the impedance of cells. Since the equalization current $I_{eq}$ supplied from the equalizer is known and constant (see Equation (2)), the voltage drop across the internal impedance of the selected cell can be determined. The open-circuit voltage of the selected cell, $V_{OC.i}$, can be estimated by compensating the voltage drop across the internal impedance $Z_{int}$, as:

$$V_{OC.i} = V_i - I_{eq}Z_{int}, \tag{5}$$

where $V_i$ is the terminal voltage of the selected cell B$_i$ ($i = 1 \dots 4$). To compensate the voltage drop $I_{eq}Z_{int}$, $Z_{int}$ needs to be measured in advance.

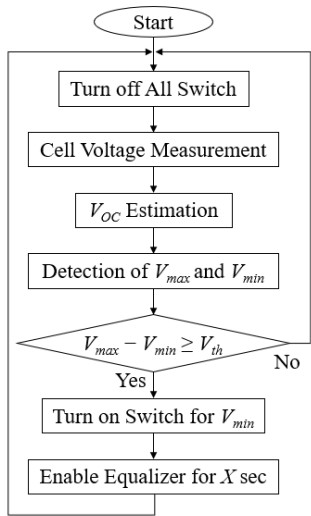

**Figure 6.** Equalization algorithm.

The most and least charged cells with the highest and lowest $V_{OC}$, $V_{max}$ and $V_{min}$, are determined. If the difference between $V_{max}$ and $V_{min}$ is less than the threshold voltage $V_{th}$, all selection switches remain off. If $V_{max} - V_{min}$ is greater than $V_{th}$, the selection switch corresponding to $V_{min}$ is turned on. Finally, Q$_H$ and Q$_L$ are driven to perform voltage equalization for X seconds. This sequence is repeated until all cell voltages are balanced, and $V_{max} - V_{min}$ will be within $V_{th}$.

A relative state of charge (SOC) compensated by the equalizer in a single cycle of the flow chart is expressed as:

$$\Delta SOC = \frac{I_{eq}X}{3600 \times C}, \tag{6}$$

where $C$ is the cell capacity in Ah. For the experimental verification test using a prototype with $I_{eq} \approx 0.75$ A for LIB cells with 3400 mAh (see Section 4.3), $X$ was determined to be 180 s so that $\Delta SOC$ was about 1%.

## 4. Experimental Results

### 4.1. Prototype

A prototype for twelve cells connected in series was built, as shown in Figure 7. Circuit elements used for the prototype are listed in Table 1. The resonant frequency $f_r$ was approximately 320 kHz, and the prototype was operated at $f_s = 120$ kHz to fulfill (1).

**Table 1.** Components list.

| Component | Value |
|---|---|
| Q$_H$, Q$_L$ | N-Ch MOSFET, FDD390N15A, $R_{on}$ = 40 m$\Omega$ |
| C$_H$, C$_L$ | Ceramic Capacitor, GRM31CB31H106KA12L 10 $\mu$F |
| C$_r$ | Film Capacitor, F161SP474M063V, 0.47 $\mu$F |
| Transformer | $N_1$:$N_2$ = 15:3, $L_{kg}$ = 13.8 $\mu$H, $L_{mg}$ = 100 $\mu$H |
| S$_1$–S$_{12}$ | N-Ch MOSFET, IRF7341PBF, $R_{on}$ = 50 m$\Omega$ |
| C$_{o1}$–C$_{o12}$ | Ceramic Capacitor, JMK325ABJ277MM-P, 220 $\mu$F |
| D$_1$–D$_{24}$ | Schottky Diode, CRS04 (T5L, TEMQ), $V_f$ = 0.49 V |

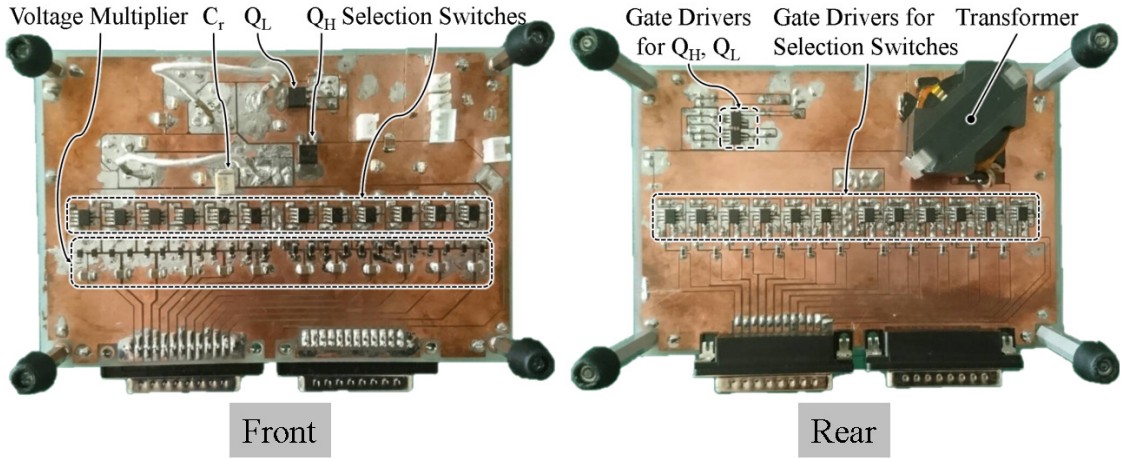

**Figure 7.** Prototype of proposed equalizer for twelve cells.

### 4.2. Measured Waveforms and Characteristics of Equalizer

Characteristics of the prototype alone were measured by breaking the node A designated in Figure 3. The equalizer was powered by an external power supply with $V_{in}$ = 48 V. All cells were removed, and a variable resistor was connected in parallel with $C_{o1}$ to emulate the current flow path in Figure 5.

The measured key operation waveforms are shown in Figure 8. These waveforms agreed well with the theoretical ones in Figure 4, verifying the operation of the prototype.

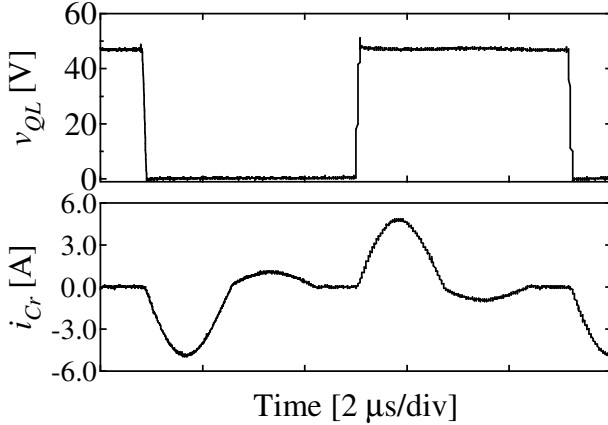

**Figure 8.** Measured key operation waveforms.

Measured output equalization current $I_{eq}$ characteristics and power conversion efficiencies are shown in Figure 9—$V_{Co1}$ in Figure 9 corresponds to the voltage of $C_{o1}$. $I_{eq}$ slightly declined with as $V_{Co1}$ increased but was nearly constant, demonstrating the constant current characteristic in DCM. The efficiency was lower than 60%, and diode forward voltage drops were considered to be the dominant loss factor as it took a significant portion of the output voltage. The measured efficiency characteristic was somewhat inferior to that of conventional equalizers (e.g., 80% [31,35]). Nevertheless, the inferior efficiency performance would be acceptable in most applications because processed power in the equalizer is one-hundredth to one-thousandth that of a main converter [36,37]. Therefore, the loss in the equalizer would be negligibly small from the system viewpoint.

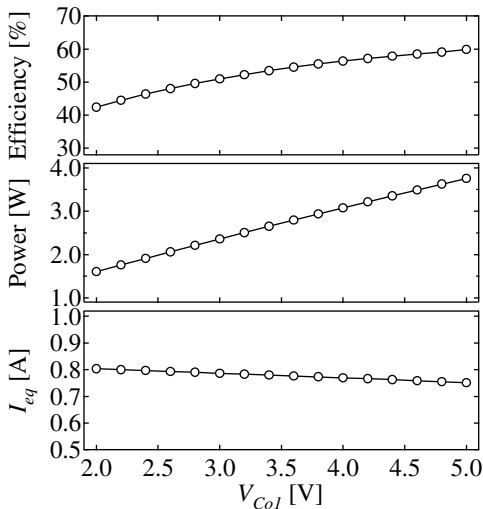

**Figure 9.** Measured output characteristics.

*4.3. Equalization Test*

An equalization test was performed for series-connected LIB cells, each with a capacity of 3400 mAh. Individual cell voltages, or SOC, were intentionally imbalanced. The experimental setup for the equalization test is shown in Figure 10. Individual cell voltages were measured using differential amplifies, and a TMS320F28335 control card was used to generate gating signals and to perform the equalization algorithm in Figure 6. The equalization was carried out with $V_{th}$ = 20 mV.

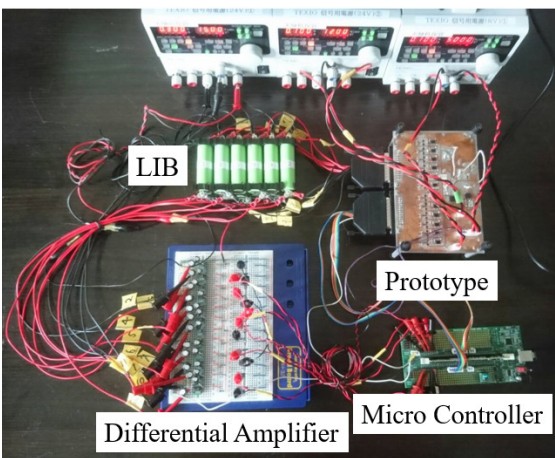

**Figure 10.** Experimental setup of equalization test for twelve LIB cells.

The resultant equalization profiles are shown in Figure 11. B$_1$ and B$_2$, the least charged cells at the beginning of the test, received an equalization current, and their voltages of $V_1$ and $V_2$ increased. At the same time, other cells supplied the input current for the equalizer, and their voltages decreased. During the course of the equalization, cells became the least charged cell alternately and received an equalization current based on the equalization algorithm. The voltage imbalance gradually vanished and decreased down to approximately 20 mV. The standard deviation of cell voltages at the end of the equalization test was as low as 10 mV, demonstrating the equalization performance of the proposed equalizer.

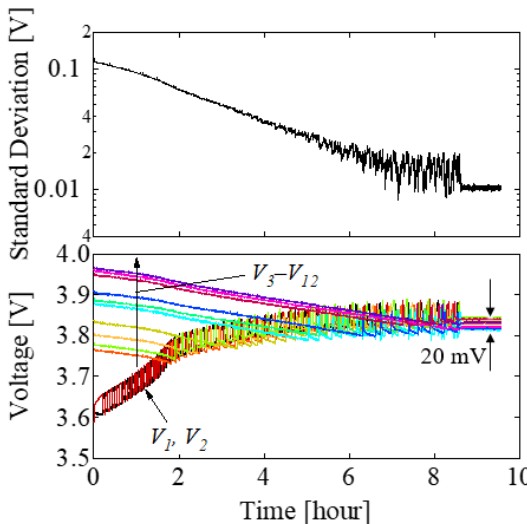

**Figure 11.** Resultant equalization profiles.

## 5. Conclusions

The cell voltage equalizer using a selective voltage multiplier for series-connected LIB cells has been proposed in this paper. The proposed equalizer can be derived by embedding selection switches into the conventional voltage multiplier-based cell voltage equalizer. In comparison with conventional cell equalizers using selection switches, the proposed topology can reduce the number of selection switches by embedding selection switched into the voltage multiplier-based equalizer, achieving the simplified circuit and reduced cost.

The equalization test using the prototype was performed for twelve LIBs connected in series. The voltage imbalance decreased to approximately 20 mV, and the standard deviation of cell voltages at the end of the equalization test was as low as 10 mV, demonstrating the equalization performance of the proposed cell equalizer.

**Author Contributions:** Conceptualization, M.U.; methodology, M.U.; simulation analysis, T.U. and K.Y.; validation, T.U.; writing—original draft preparation, M.U.; writing—review and editing, M.U.; supervision, M.U.

**Funding:** This research received no external funding.

**Conflicts of Interest:** The authors declare no conflict of interest.

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
