# Peer review of "Cell Voltage Equalizer Using a Selective Voltage Multiplier with a Reduced Selection Switch Count for Series-Connected Energy Storage Cells"

_electronics, doi:10.3390/electronics8111303_

Round 1

Reviewer 1 Report

This paper presents a new topology for voltage balancing of cells in a Supercapacitor module. The novelty level of this paper is average. There are the following issues that should be considered to improve the paper quality:
1- In the abstract and introduction, please explain how and why the proposed topology reduces the number of selection switches?
2-You should talk about figures in order. The authors can not mention figure 2 (line 41) before figure 1.
3- Please add some references for each topology in Figure 2
4- To realize the ideal switch in Figure 2, only a switch like Mosfet is enough, but we need minimum two switches for realization in Figure 3. Please talk more realistic about the number of switches. I think the statement within lines 88 and 93 should be reconsidered.
5-Please present more results about charging and discharging states of the supercapacitor.

Author Response

We are very grateful to you for your thoughtful and helpful review of the manuscript. Your comments and suggestions have been incorporated as appropriate into the revised manuscript. The revised parts are highlighted with green in the revised manuscript. Please see the attched file.

Reviewer 2 Report

In this study, a pack-to-cell battery equalizer based on voltage multiplier topology is proposed. The main contribution of this paper is to integrate the selection switch (bidirectional) with the voltage multiplier. Hence, the number of selection switches can be reduced. According to the experimental results, the proposed equalizer can reduce the imbalance to about 20 mV. These results are adequate; however, some modifications are required to make this paper more complete.

By integrating the selection switch (bidirectional) with the voltage multiplier, the number of selection switches can be reduced from n+5 to n. However, an additional voltage multiplier is required (comparing with Fig. 2(b)). Please simply comment on this issue (pros and cons, suitable application vs Fig. 2(b)) in the introduction section. Please explain the “Voc estimation” in Fig. 6, also discuss how to determine the value of “X” sec in Fig. 6. Please compare the experimental results of your method to some existing technology. The efficiency shown in Fig. 9 is somewhat low for a topology using a resonant inverter, please simply comment on this issue. Please provide more experimental results using different initial conditions (simulation or experimental).

Author Response

(The authors gave the same response as above.)

Round 2

Reviewer 1 Report

This version of the paper is accepted for publication. 

Reviewer 2 Report

The authors have made all the required revisions